# EventFlash: Towards Efficient MLLMs for Event-Based Vision

**Shaoyu Liu**[1,2], **Jianing Li**[2*], **Guanghui Zhao**[1*], **Yunjian Zhang**[2], **Wen Jiang**[3],
**Ming Li**[4*], **Xiangyang Ji**[2]
[1] Xidian University, [2] Tsinghua University, [3] Beijing Institute of Technology,
[4] Guangdong Laboratory of Artificial Intelligence and Digital Economy (SZ)
`liusy@stu.xidian.edu.cn, ghzhao@xidian.edu.cn`

## Abstract

Event-based multimodal large language models (MLLMs) enable robust perception in high-speed and low-light scenarios, addressing key limitations of frame-based MLLMs. However, current event-based MLLMs often rely on dense image-like processing paradigms, overlooking the spatiotemporal sparsity of event streams and resulting in high computational cost. In this paper, we propose EventFlash, a novel, efficient MLLM to explore spatiotemporal token sparsification for reducing data redundancy and accelerating inference. Technically, we built EventMind, a large-scale and scene-diverse dataset with over 500k instruction sets, providing both short and long event stream sequences to support our curriculum training strategy. Then, we present the adaptive temporal window aggregation module for efficient temporal sampling, which adaptively compresses temporal tokens while retaining key temporal cues. Finally, the sparse density-guided attention module is designed to improve spatial token efficiency by selecting informative regions and suppressing empty or sparse areas. Experimental results show that EventFlash achieves a $12.4\times$ throughput improvement over the baseline (EventFlash-Zero) while maintaining comparable performance. It supports long-range event stream processing with up to 1,000 bins, significantly outperforming EventGPT's 5-bin limit. We believe EventFlash serves as an efficient foundation model for event-based vision. Our codes will be released in `https://github.com/XduSyL/EventFlash`

## 1 Introduction

Event cameras (Gallego et al., 2020; Posch et al., 2014; Li & Tian, 2021), bio-inspired vision sensors, operate differently from frame-based cameras. Each pixel responds to intensity changes by generating asynchronous events (Li et al., 2022; Kudithipudi et al., 2025). Due to their high temporal resolution and high dynamic range, event cameras have been applied to various vision tasks (e.g., scene understanding (Zhu et al., 2018; Kong et al., 2024; Yao et al., 2024; Zhou et al., 2024; Li et al., 2025a; Liu et al., 2025b; Li et al., 2025b; Zhou & Lee, 2025)) in high-speed or low-light scenarios.

Recent multimodal large language models (MLLMs) (Xiang et al., 2025; Li et al., 2024a; Tang et al., 2025; Huang et al., 2024a; Qian et al., 2024; Fang et al., 2026a;b; 2024a; Xu et al., 2024; Yan et al., 2025b) have achieved remarkable breakthroughs in processing conventional frames and language, showing strong capabilities in scene understanding and visual question answering. However, these models are primarily designed for frame-based inputs and cannot directly handle the unique spatiotemporal properties of event streams. A straightforward approach to extending MLLMs to event-based vision involves converting event streams into dense, image-like representations before feeding them into existing MLLMs (e.g., LLaVA (Liu et al., 2023), GPT-4 (Bubeck et al., 2023), or Qwen (Bai et al., 2023)). However, this transformation often overlooks the inherent spatiotemporal sparsity of event data and introduces substantial redundancy (Gehrig & Scaramuzza, 2024; Messikommer et al., 2020; Wu et al., 2024a). In other words, applying dense image-like processing paradigms to event streams not only incurs significant computational overhead but also substantially limits the effective

---

*Corresponding authors

length and efficiency of event stream understanding. Thus, developing efficient MLLMs that fully exploit the unique spatiotemporal properties of event data remains a critical and unresolved challenge.

Despite recent progress, most existing event-based MLLMs (Liu et al., 2025b; Li et al., 2025b; Zhou & Lee, 2025; Liu et al., 2025a) still rely on dense image-like representations, which hinders computational efficiency and scalability to long event sequences. For example, EventGPT (Liu et al., 2025b) converts event streams into dense token sequences for language modeling. EventVL (Li et al., 2025b) integrates RGB frames with event data to enhance multimodal reasoning. LLaFEA (Zhou & Lee, 2025) employs frame-event fusion for region-level spatiotemporal grounding. Although these models perform well in challenging scenarios such as high-speed motion and low-light conditions, their dense processing of sparse event data leads to significant overhead and limits real-time or long-range understanding. Meanwhile, the scene diversity of their datasets is relatively limited, and the event streams are short, making it difficult to support general-purpose models for long event-stream understanding.

In this paper, we propose EventFlash, a novel efficient MLLM that leverages spatiotemporal token sparsification to reduce data redundancy and accelerate inference. Unlike prior works that focus on maximizing reasoning accuracy, our goal is to address three key challenges in efficient MLLMs: (i) *Temporal inefficiency*: The microsecond resolution of event streams results in prohibitively large token volumes when processed over long temporal durations; (ii) *Spatial inefficiency*: The inherent sparsity of event data leads to numerous empty or low-information tokens that incur computational overhead due to uniform attention allocation; (iii) *Dataset limitations*: Existing instruction-augmented datasets are not publicly available and often lack diversity, contain low-quality annotations, and cover short temporal sequences, making them inadequate for training generalizable models.

To address these challenges, our EventFlash presents a density-aware spatiotemporal token sparsification strategy that exploits the inherent sparsity and high temporal resolution of event streams. Specifically, we propose an adaptive temporal window aggregation module for efficient temporal sampling, which dynamically compresses temporal tokens while preserving essential temporal cues. Then, a sparse density-guided attention module is presented to enhance spatial token efficiency by selecting informative regions and suppressing empty or low-density areas. Moreover, we design a progressive curriculum learning strategy following a short-to-long paradigm to improve EventFlash's generalization and generative capabilities. To support this, we built a large-scale scene-diverse dataset over 500k instruction sets, including both short and long event stream sequences. Experimental results show that EventFlash achieves a 12.4× improvement in throughput over our baseline (EventFlash-Zero) while maintaining comparable performance. Notably, EventFlash enables long-range event stream processing of up to 1,000 bins compared to only 5 bins in the competing EventGPT.

In summary, the main contributions of this work are:

- We propose EventFlash, ***an efficient event-based vision MLLM***, which explores a spatiotemporal token sparsification strategy for raw event streams to reduce redundancy, accelerate inference (12.4× throughput), and enable long-range event stream understanding (up to 1,000 bins).
- We present a *density-aware spatiotemporal token sparsification* strategy for event-based MLLMs, which effectively reduces redundancy while maintaining comparable reasoning accuracy by leveraging the fine-grained temporal resolution and inherent sparsity of raw event streams.
- We build a *large-scale scene-diverse dataset for long-range event stream understanding*. We believe this standardized dataset will accelerate future research in event-based MLLMs.

## 2 RELATED WORK

**Event-based Vision with MLLMs.** Early works (Wu et al., 2023; Zhou et al., 2023) have explored the alignment between event data and textual information. Event-CLIP (Wu et al., 2023) builds on pre-trained vision-language models (Radford et al., 2021; Yang et al., 2023; Klenk et al., 2024; Huang et al., 2024b; Tong et al., 2025) for event-based recognition, and EventBind (Zhou et al., 2023) incorporates an event encoder to unify images, events, and texts. Yet both overlook the world knowledge embedded in LLMs, constraining nuanced scene understanding. More recently, emerging event-based MLLMs (Liu et al., 2025b; Li et al., 2025b; Zhou & Lee, 2025) have demonstrated strong reasoning capabilities in challenging conditions. For example, EventGPT (Liu et al., 2025b) is the first to design an event-based MLLM for accurate generation. EventVL (Li et al., 2025b) enhances

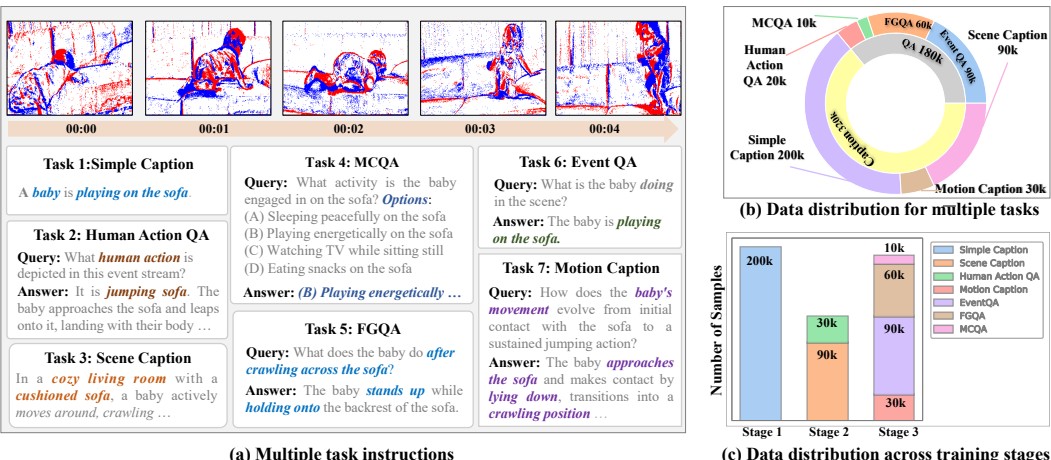

Figure 1: Instructions and data statistics of our EventMind. (a) Seven tasks instructions for event stream understanding. (b) Data distributions of each task. (c) Data distributions of the three stages.

robustness by fusing complementary modalities from event streams and RGB frames. LLaFEA (Zhou & Lee, 2025) achieves region-level spatiotemporal grounding through the complementary fusion of frame and event modalities. However, these event-based LLMs rely on dense image-like processing of inherently sparse events (Peng et al., 2024; Perot et al., 2020; Vemprala et al., 2021; Zhu et al., 2022; Tulyakov et al., 2019; Qu et al., 2024; Lin et al., 2023; Shrestha & Orchard, 2018; Wu et al., 2024b; Engelken, 2023; Cho et al., 2024; Wan et al., 2022; Mei et al., 2023; Li et al., 2025c; 2023; Chen et al., 2024), leading to excessive computation and hindering long-sequence inference.

**Efficient Token Sparsification in MLLMs.** Recent MLLMs (Weng et al., 2024; Jiang et al., 2025; Qian et al., 2024; Yan et al., 2025a) have revealed that visual tokens extracted from foundation models like CLIP contain substantial redundancy, leading to significant computational overhead. Consequently, several token sparsification strategies (Yehezkel et al., 2024; He et al., 2024; Zhang et al., 2024) have been attempted to reduce token counts while preserving essential semantics in video tasks. *However, asynchronous events differ fundamentally from structured frames: while video redundancy mainly stems from spatial repetition within a regular patch grid, event streams consist of sparse spatiotemporal points with redundancy arising from uneven temporal sampling. Their tokens are distributed irregularly and vary in density, making frame-based sparsification not only computationally costly but also ineffective for long event stream understanding.* Thus, this work presents a novel spatiotemporal token sparsification strategy specifically tailored for event streams.

# 3 EVENTMIND DATASET

**Data Collection**. To support the curriculum learning strategy in our EventFlash, we construct a large-scale multimodal dataset named EventMind for event stream understanding. EventMind provides long temporal sequences, diverse scenes, multiple tasks, and high-quality instructions. The raw event data is sourced from both real-world and synthetic domains. Real-world data includes short-duration event sequences from DSEC (Gehrig et al., 2021) and N-ImageNet (Kim et al., 2021), as well as longer-duration streams from HARDVS (Wang et al., 2024b) and E2VID (Rebecq et al., 2019). Synthetic data are generated by converting large-scale video datasets (i.e., Kinetics-700 (Carreira et al., 2019), UCF-101 (Soomro et al., 2012), Wevid-10 M (Bain et al., 2021), PLM-Data (Cho et al., 2025), and MotionBench (Hong et al., 2025)) into event streams using the V2E simulator (Hu et al., 2021). To ensure high-quality simulated events, we use GPT-4o to automatically filter videos using their captions before simulation. To align with our curriculum stages, we categorize them into three groups: short (0–50 ms), medium (50–5,000 ms), and long (5,000–20,000 ms).

**Instruction Generation**. To evaluate the modeling capacity and generalization ability of our EventFlash, we define seven distinct task types for event stream understanding. As shown in Fig. 1(a), these tasks include motion captioning, event question answering (Event QA), human action QA, multiple-choice QA (MCQA), simple captioning, fine-grained QA (FGQA), and scene captioning.

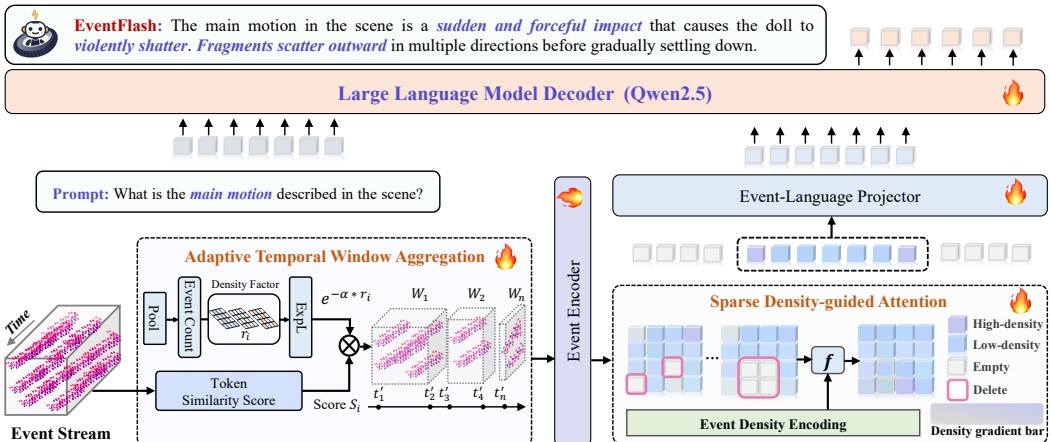

Figure 2: The pipeline of efficient MLLMs (**EventFlash**). The adaptive temporal window aggregation module is presented for efficient temporal sampling, which adaptively compresses temporal tokens while retaining key temporal cues. Besides, the sparse density-guided attention module is designed to improve spatial token by selecting informative regions and suppressing empty or sparse areas.

Text instructions are constructed via two pathways: (i) For samples with existing textual annotations, we use GPT-4o to refine the descriptions by removing static attributes and irrelevant visual details (e.g., texture, color), ensuring better alignment with event streams. (ii) For samples lacking ground-truth text, we leverage Qwen-VL-Max to automatically generate annotations from corresponding video inputs, enabling a scalable and consistent data synthesis pipeline. In addition, we organize a multi-person team to manually inspect and filter the generated instruction sets for quality assurance.

**Dataset Statistics**. We analyze the composition of the EventMind dataset from a curriculum learning perspective (see Fig. 1(b)). It is structured into three stages based on event sequence length and task complexity. In Stage 1, short sequences are used for the simple captioning task, contributing 200k instruction samples. Stage 2 utilizes medium-length sequences for scene captioning and human action understanding, with a total of 110k instructions. Stage 3 focuses on long sequences for more complex tasks such as motion captioning, EventQA, FGQA, and MCQA, comprising 190k instructions. Overall, our EventMind comprises 500k instruction samples spanning seven task types (see Fig. 1(c)): 200k for simple captioning, 90k for scene captioning, 30k for motion captioning, 90k for EventQA, 60k for FGQA, 10k for MCQA, and 20k for human action QA.

All in all, the novel event-text modality and labor-intensive design make EventMind a highly competitive dataset with several key strengths: (i) *High temporal sampling resolution at the microsecond level from event streams*; (ii) *Coverage of temporal sequences of various lengths*; (iii) *Diverse scene types supporting 7 distinct tasks*; (iv) *A large-scale high-quality instruction set with 500k samples*.

## 4 METHOD

### 4.1 EVENTFLASH OVERVIEW

This work aims at designing an efficient MLLM for event stream understanding, termed EventFlash, which presents a spatiotemporal token sparsification strategy to reduce redundancy and accelerate inference. As illustrated in Fig. 2, our framework consists of five modules: ***adaptive temporal window aggregation module***, ***sparse density-guided attention module***, event encoder, event-language projector, and large language model (LLM) decoder. More precisely, the adaptive temporal window aggregation module first segments the continuous event stream into uniform short bins and adaptively merges adjacent bins based on token similarity or event density. These processed bins are then passed by an event encoder (e.g., CLIP) to extract semantic embeddings. In parallel, the sparse density-guided attention module improves spatial token efficiency by emphasizing informative regions and suppressing empty or low-density areas. The event-language projector aligns the event tokens with text tokens to enable coherent multimodal fusion. Finally, the compact event tokens are fused with text tokens and processed by an LLM decoder (e.g., Qwen-2.5) for multimodal generation tasks.

## 4.2 TEMPORAL SPARSE

The microsecond-level resolution of raw event streams generates an excessive number of temporal tokens, resulting in high computational overhead. To address this, we introduce a two-stage density-guided adaptive temporal window aggregation (ATWA) module that compresses event streams while preserving key motion dynamics. The event stream is first divided into fine-grained bins, which are iteratively merged based on an asynchronous spatiotemporal spike metric (Li et al., 2022). Each bin is treated as a polarity-aware spatiotemporal point process with an intensity function $\lambda_B$:

$$\lambda_B(x, y, t, p) = \sum_{e_n \in B} f(p_n) \cdot \exp\left(-\frac{(x - x_n)^2}{2\sigma_x^2} - \frac{(y - y_n)^2}{2\sigma_y^2} - \frac{(t - t_n)^2}{2\sigma_t^2}\right), \tag{1}$$

where $f(p_n)$ encodes the polarity for an event $(x_n, y_n, t_n, p_n)$. $\sigma_x$, $\sigma_y$, and $\sigma_z$ are the parameters of the Gaussian kernel. The similarity distance between two bins $B_i$ and $B_{i+1}$ can be computed as:

$$D(B_i, B_{i+1}) = \left\|\lambda_{B_i} - \lambda_{B_{i+1}}\right\|_2, \tag{2}$$

where a lower $D$ indicates higher temporal correlation between two bins. We iteratively merge adjacent bins when the distance is below a threshold $\tau$, forming meta event windows $\{M_1, M_2, \ldots, M_K\}$.

In the second stage, we perform semantic-aware aggregation of meta bins. Each window $M_i$ is passed through an event encoder (e.g., ViT (Arnab et al., 2021)) to obtain a CLS token representation $z_i$, and the similarity $S_i$ between adjacent windows is defined as cosine similarity as follows:

$$S_i = \frac{z_i^\top z_{i+1}}{\|z_i\| \cdot \|z_{i+1}\|}. \tag{3}$$

To incorporate event sparsity, we define a normalized event density factor $r_i = \frac{1}{|M_i|} \sum_{e_n \in M_i} \mathbf{1}_{e_n}$, and compute a density-aware weight. The final adaptive merging score can be formulated by:

$$A_i = S_i \cdot \exp(-\alpha \cdot r_i), \tag{4}$$

where $\alpha$ controls the decay sensitivity. which jointly considers semantic similarity and event sparsity. We iteratively merge windows with high $A_i$ to obtain a compressed yet semantically meaningful temporal sequence that preserves key temporal cues with reduced computational cost.

## 4.3 SPATIAL SPARSE

While temporal aggregation reduces sequence length, spatial redundancy still persists due to the inherent sparsity and uneven event distribution across the sensor plane. To tackle this, we propose the sparse density-guided attention (SDGA) module (see Fig. 3), which adaptively prunes uninformative tokens based on both visual semantics and event density. For each aggregated event bin, we use an encoder (i.e., ViT) to extract patch-level features $\{x_j\}_{j=1}^n$, which are fed into a multi-head self-attention mechanism as:

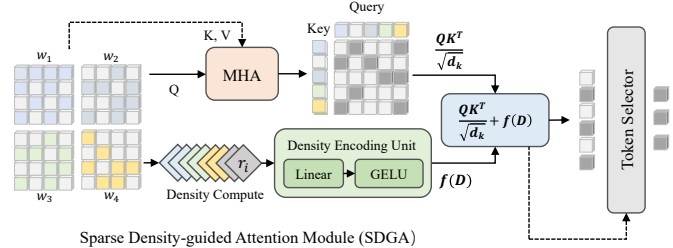

Figure 3: The architecture of the sparse density-guided attention module. It enhances spatial token efficiency by selecting informative regions and suppressing empty or low-density areas.

$$\text{Attention}(Q, K, V) = \text{softmax}\left(\frac{QK^\top}{\sqrt{d_k}}\right)V, \tag{5}$$

where $Q$, $K$, and $V$ are the projected queries, keys, and values from $\{x_j\}$, and $d_k$ is the key dimension.

In parallel, we compute the event density $D_j$ of each token region based on the number of events falling within its receptive field. This scalar value is then passed through a density encoding unit consisting of a linear transformation followed by GELU activation:

$$f(D_j) = \text{GELU}(\text{Linear}(D_j)), \tag{6}$$

where $f(D_j)$ is a soft modulation signal that reflects the importance of each spatial token. The encoded density is added to the attention scores to focus on denser and more important areas as:

$$\tilde{A}_{ij} = \frac{Q_i K_j^\top}{\sqrt{d_k}} + f(D_j). \tag{7}$$

Finally, we apply a *Token Selector* operation that ranks the aggregated attention responses and discards low-importance tokens, which can be formulated as follows:

$$\hat{x}_i = \text{TokenSelector}\left(\sum_j \text{softmax}(\tilde{A}_{ij}) \cdot V_j\right). \tag{8}$$

In summary, this density-guided token pruning strategy enables EventFlash to keep important spatial details while greatly cutting down on redundant computations. By combining semantic relevance with event density, SDGA produces more compact tokens for the efficient MLLM.

### 4.4 Short-to-Long Curriculum Learning

To support scalable training across different event durations and enhance generalization, we propose a progressive short-to-long curriculum learning strategy. Unlike prior event-based MLLMs such as EventVL (Li et al., 2025b) and EventGPT (Liu et al., 2025b), which train different modules in separate stages, our curriculum emphasizes a gradual progression from short to long event streams. This design facilitates smoother training dynamics, enabling EventFlash to evolve from mastering simple alignments to handling complex reasoning and long-range event understanding.

To be specific, **Stage 1** focuses on event-language alignment by training on 200k short sequences (0-50 ms) paired with simple scene descriptions to establish basic cross-modal understanding. **Stage 2** expands to 110k medium sequences (50-5,000 ms) featuring complex motions like human actions, enhancing the model's reasoning and ability to handle instruction-following and event-based QA over longer inputs. **Stage 3** fine-tunes the model on 190k long sequences (5,000–20,000 ms) with rich scene descriptions, enabling holistic scene understanding and open-ended language generation.

## 5 Experiments

### 5.1 Experimental Setup

**Implements Details.** We initialize the event encoder with CLIP-ViT-Large-Patch14 (Radford et al., 2021) and use Qwen2.5 (Bai et al., 2023) as the LLM backbone. A two-layer MLP serves as the Event-Language Projector to align the event and semantic spaces. EventFlash is implemented in both 3B and 7B variants and trained on 8 A100 GPUs. For throughput evaluation, the inference is conducted on an A100 GPU using Hugging Face deployment. Our three-stage curriculum learning strategy proceeds as follows: only the Event-Language alignment module is trained in Stage 1, using a learning rate of $2 \times 10^{-3}$ and a batch size of 64. For Stage 2 and Stage 3, all model parameters are unfrozen and trained with a learning rate of $2 \times 10^{-5}$, a batch size of 8, and a gradient accumulation step of 4. A cosine learning rate decay schedule is applied throughout training. We set the temporal aggregation interval to 10 ms and use a density attenuation factor $\alpha$ of 0.1 for spatial sparsification.

**Evaluation Metrics.** To thoroughly evaluate the generalization and reasoning capabilities of our EventFlash, we adopt four metrics aligned with protocols established in LLaVA (Liu et al., 2023) and other widely used benchmarks (Fang et al., 2024b). More precisely, we use the following evaluation metrics: (i) Global detailed captioning (GDC) to assess scene-level summarization, (ii) Fine-grained question answering (FGQA) to evaluate the model's understanding of localized event details, (iii) Human action question answering (HAQA) to measure temporal reasoning at the action level, and (iv) Multiple choice question answering (MCQA) to assess instruction-following and discriminative reasoning. For open-ended tasks (GDC and FGQA), we employ LLM-based evaluation using GPT-4o (i.e., LLM-Judge) consistent with prior benchmarks. For HAQA and MCQA, we report the accuracy based on exact matches with ground-truth answers. In addition, throughput and maximum event bin capacity are used to evaluate the efficiency of all MLLMs. Throughput is typically defined as the number of tokens generated per second during inference, while maximum event bin capacity refers to the largest number of event bins the model can process in a single input.

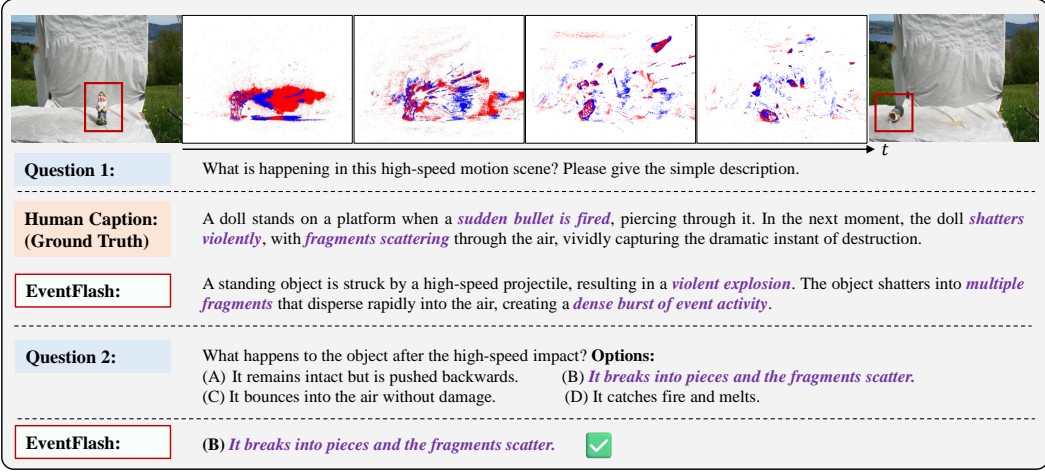

Figure 4: Representative visualization tests on motion captioning and multiple-choice question answering (MCQA) are conducted in high-speed scenarios. Our EventFlash demonstrates superior accuracy in recognizing fast-moving objects, such as a sudden bullet being fired at a doll.

## 5.2 QUALITATIVE RESULTS

Table 1: Comparison of video-based MLLMs and event-based MLLMs on our EventMind dataset and EventChat-Sub dataset (Liu et al., 2025b). Notably, it can process significantly longer event bins than the event-based competitor EventGPT.

| Models | Params | LLM Backbone | Max Bins | Throughput (Token/s) | EventMind | | | | EventChat-Sub | |
|---|---|---|---|---|---|---|---|---|---|---|
| | | | | | GDC | FGQA | HAQA | MCQA | GDC | FGQA |
| *Video-Base ∼3B Scale MLLMs* | | | | | | | | | | |
| **Qwen2.5 VL** (Bai et al., 2023) | 3B | Qwen2.5 | 768 | – | 20.6 | 41.7 | 23.8 | 34.6 | 34.5 | 51.2 |
| **VideoChat2-Flash** (Li et al., 2024b) | 2B | Qwen2.5 | 1,000 | – | 31.6 | 38.9 | 16.2 | 43.6 | 36.9 | 43.8 |
| **InternVL2.5** (Lu et al., 2025) | 4B | Qwen2.5 | – | – | 17.9 | 37.0 | 21.3 | 27.3 | 28.9 | 44.6 |
| *Video-Base ∼7B Scale MLLMs* | | | | | | | | | | |
| **VideoChat2-Flash** (Li et al., 2024b) | 7B | Qwen2.5 | 1,000 | – | 36.2 | 41.9 | 18.9 | 48.2 | 53.1 | 53.6 |
| **LLaVA-Next-Video** (Liu et al., 2023) | 7B | Qwen2.5 | 56 | – | 31.2 | 44.6 | 22.8 | 42.7 | 46.3 | 54.8 |
| **Qwen2.5 VL** (Bai et al., 2023) | 7B | Qwen2.5 | 768 | – | 22.1 | 43.9 | 28.6 | 41.8 | 41.6 | 53.2 |
| **InternVL2.5** (Lu et al., 2025) | 8B | InternLM2.5 | – | – | 19.7 | 40.0 | 25.3 | 38.2 | 42.5 | 55.6 |
| *Event-Base MLLMs* | | | | | | | | | | |
| **EventGPT-7B** (Liu et al., 2025b) | 7B | Vicuna-v1.5 | 5 | 42.2 | – | – | – | – | 71.2 | 78.2 |
| **EventFlash-Zero** | 3B | Qwen-2.5 | **1,000** | 2.3 | 45.3 | 60.4 | 85.0 | 58.2 | 70.4 | 77.1 |
| 🦦 **EventFlash-3B (Ours*)** | 3B | Qwen-2.5 | **1,000** | 28.5 | 46.8 | 61.1 | 84.9 | 60.0 | 71.5 | 78.6 |
| 🦦 **EventFlash-7B (Ours*)** | 7B | Qwen-2.5 | **1,000** | 24.0 | **52.3** | **64.2** | **87.6** | **63.1** | **74.1** | **79.5** |

**Comparison with State-of-the-Art MLLMs.** To evaluate the effectiveness and efficiency of EventFlash, we compare it against four state-of-the-art video-based MLLMs and the only open-sourced event-based MLLM (i.e., EventGPT (Liu et al., 2025b)). We select strong video-based models at both the 3B and 7B scales, including Qwen2.5-VL (Bai et al., 2023), VideoChat2-Flash (Li et al., 2024b), LLaVA-Next-Video (Liu et al., 2023), and InternVL 2.5 (Lu et al., 2025). EventGPT uses fixed bin encoding for event stream understanding. We also construct a baseline, EventFlash-Zero, by removing spatiotemporal sparsification from EventFlash.

**Qualitative Evaluation.** As illustrated in Table 1, EventFlash outperforms four video-based MLLMs and the event-based EventGPT on all four tasks (i.e., GDC, FGQA, HAQA, and MCQA). This demonstrates that EventFlash excels at understanding and describing dynamic event scenes. While EventGPT implements a fixed configuration of 5 event bins, EventFlash can process up to 1,000 event bins, achieving a 200× increase in processing capacity. In other words, our EventFlash is enabled by our efficient sparsification strategy for longer-term understanding. In addition, EventFlash reaches a speed of 28.5 tokens per second during inference. This is 12.4× faster than our baseline EventFlash-Zero (2.3 tokens per second), and it still maintains comparable performance on all tasks.

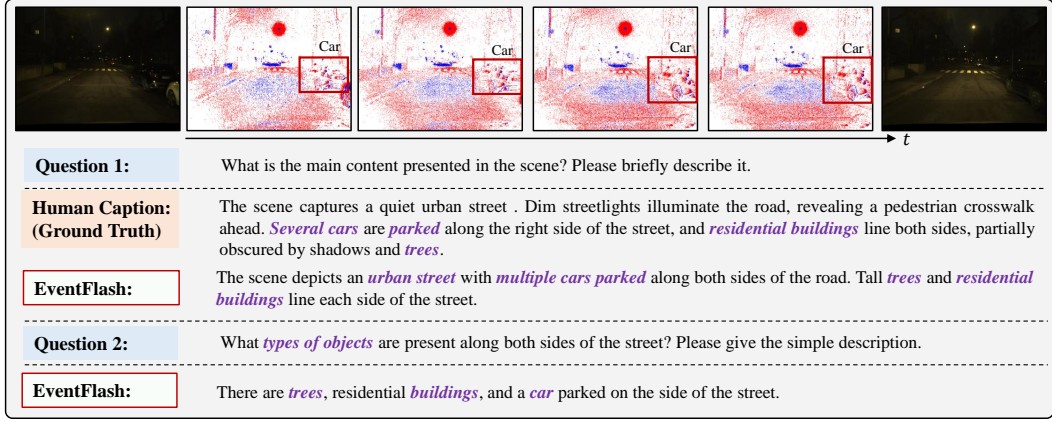

| | |
|---|---|
| **Question 1:** | What is the main content presented in the scene? Please briefly describe it. |
| **Human Caption:** (Ground Truth) | The scene captures a quiet urban street . Dim streetlights illuminate the road, revealing a pedestrian crosswalk ahead. *Several cars* are *parked* along the right side of the street, and *residential buildings* line both sides, partially obscured by shadows and *trees*. |
| **EventFlash:** | The scene depicts an *urban street* with *multiple cars parked* along both sides of the road. Tall *trees* and *residential buildings* line each side of the street. |
| **Question 2:** | What *types of objects* are present along both sides of the street? Please give the simple description. |
| **EventFlash:** | There are *trees*, residential *buildings*, and a *car* parked on the side of the street. |

Figure 5: Representative visualization tests on event questioning answering (QA) and scene caption are conducted in low-light scenarios. EventFlash showcases strong scene description and reasoning capabilities, such as identifying a car in a nighttime scene where it is barely visible on RGB images.

**Visualization Evaluation.** We further evaluate EventFlash under challenging scenarios, such as high-speed motion and low illumination. As shown in Fig. 4 and Fig. 5, our model demonstrates strong descriptive and reasoning capabilities in both cases. *In high-speed case:* The scene depicts a goblin being struck by a high-velocity projectile, resulting in a mid-air explosion with scattered fragments. EventFlash generates an accurate and fine-grained description of this dynamic event and correctly answers a multiple-choice question. *In low-light case:* The scenario involves a vehicle driving through darkness. Despite the absence of frame-based visual cues, EventFlash generates a coherent and precise description, along with an accurate response to the corresponding QA prompt. These results validate EventFlash's ability to understand complex dynamics in edge-case environments where traditional frame-based models often fail.

To further demonstrate the advantages of EventFlash on long-duration event streams, we compare it with EventGPT on a 10,000 ms sequence. As shown in Fig. 6, EventGPT operates on a fixed number of bins (e.g., 0–50 ms), limiting its understanding to moment-level segments. In contrast, EventFlash leverages its high maximum event bin capacity to process extended sequences, enabling coherent reasoning across the full temporal window and capturing sequence-level motion dynamics. As a result, EventFlash generates more contextually accurate de-

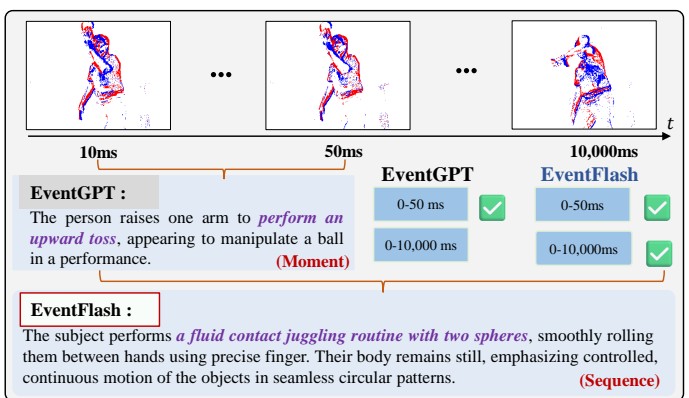

Figure 6: Comparison of EventFlash and EventGPT on long-duration event streams from our EventMind dataset.

scriptions, highlighting its potential for real-world applications that require long-range understanding, such as surveillance analysis and autonomous driving.

This gap stems from their different temporal modeling strategies. EventGPT relies on short, fixed-duration bins, which fragment long-term motion and hinder the capture of cross-bin dependencies. In contrast, EventFlash maintains a unified representation over extended event streams, preserving temporal continuity and enabling consistent reasoning across long time horizons. As a result, EventFlash produces descriptions that better reflect the overall motion evolution of the scene, rather than isolated moment-level observations.

## 5.3 ABLATION STUDY

**Contribution of Each Component**. To explore the impact of each component on overall performance, we conduct an ablation study by comparing our full model against three variants: a baseline without any sparsification (EventFlash-Zero), a model with only temporal sparsification, and a model with only spatial sparsification. As shown in Table 2, our full model achieves a 12.4× increase in throughput (28.5 tokens/s vs. 2.3 tokens/s) while maintaining comparable performance across four evaluation metrics (i.e., GDC, FGQA, HAQA, and MCQA). With temporal sparsification alone, the model achieves 14.0 tokens/s, representing a 6.1× speedup over the baseline. In contrast, spatial sparsification alone yields a 2.3× improvement, reaching 5.3 tokens/s. The results show that both temporal and spatial sparsification contribute to efficiency gains.

Table 2: The contribution of each component to our EventMind dataset. The baseline uses our EventFlash without the spatiotemporal token sparsification strategy.

| Model | S | T | Token/s | EventMind | | | |
|---|---|---|---|---|---|---|---|
| | | | | GDC | FGQA | HAQA | MCQA |
| Baseline | ✗ | ✗ | 2.3 | 45.3 | 60.4 | 85.0 | 58.2 |
| A | ✓ | ✗ | $5.3_{+2.3\times}$ | 46.3 | 61.2 | 85.1 | 59.6 |
| B | ✗ | ✓ | $14.0_{+6.1\times}$ | 47.1 | 60.6 | 83.8 | 60.3 |
| **Ours*** | ✓ | ✓ | $28.5_{+12.4\times}$ | 46.8 | 61.1 | 84.9 | 60.0 |

**Influence of the Aggregation Interval Length**. To explore how the initial temporal bin duration affects performance and efficiency, we evaluate model throughput and accuracy across different initial event bin durations. As shown in Table 3, we compare four settings with bin lengths of 5 ms, 10 ms, 20 ms, and 30 ms. We observe that shorter bin durations (e.g., 5 ms) provide finer temporal resolution but significantly increase the number of windows, resulting in lower throughput (15.8 tokens/s) compared to our default setting of 28.5 tokens/s at 10 ms. Despite the increased computational load, the model maintains strong performance across all tasks. Conversely, increasing the bin size to 20 ms and 30 ms improves throughput to 52.6 and 63.3 tokens/s, respectively, indicating greater efficiency. However, this comes at the cost of performance degradation on GDC, FGQA, MCQA, and HAQA. In this work, a bin duration of 10 ms offers a trade-off between accuracy and efficiency, and is therefore adopted as our default setting.

Table 3: The influence of aggregation interval length on our EventMind dataset.

| Aggregation interval | Throughput (Token/s) | EventMind | | | |
|---|---|---|---|---|---|
| | | GDC | FGQA | MCQA | HAQA |
| 5ms | 15.8 | 47.1 | **61.8** | 84.6 | 58.2 |
| 10ms | 28.5 | **46.8** | 61.1 | **84.9** | **60.0** |
| 20ms | 52.6 | 43.2 | 56.3 | 72.6 | 48.4 |
| 30ms | 63.3 | 36.8 | 48.2 | 61.8 | 46.2 |

**Impact of Density Attenuation Factor $\alpha$.** We investigate how the density attenuation factor $\alpha$ affects model throughput and task performance (see Table 4). To explore the trade-off between density-guided and similarity-guided token merging, we evaluate four values of $\alpha$ to identify the optimal balance between accuracy and efficiency. The results show that increasing $\alpha$ leads to higher throughput, indicating that stronger density suppression accelerates the token aggregation process. For example, FGQA and MCQA stay mostly stable when $\alpha$ is between 0.2 and 0.4. However, GDC and HAQA rely more on detailed timing information. Because of this, their performance drops when $\alpha$ gets higher. The results confirm the effectiveness of our density-aware weighting mechanism. Notably, $\alpha = 0.1$ and $\alpha = 0.4$ achieve a favorable trade-off, providing substantial speed gains while preserving strong task performance.

Table 4: The influence of density factor $\alpha$ on throughput and performance on our EventMind dataset.

| Density Factor $\alpha$ | Throughput (Token/s) | GDC | FGQA | MCQA | HAQA |
|---|---|---|---|---|---|
| 0.1 | 28.5 | 46.8 | 61.1 | 84.9 | 60.0 |
| 0.2 | 27.6 | 45.6 | 61.4 | 85.2 | 58.4 |
| 0.4 | 28.8 | 45.3 | 61.6 | 85.2 | 58.4 |
| 0.6 | 26.8 | 47.2 | 60.8 | 83.2 | 60.1 |

## 5.4 EXTENSIVE APPLICATION

We further investigate additional downstream applications enabled by our EventFlash. For instance, EventFlash can be readily fine-tuned to support action recognition tasks. As shown in 5, we evaluate its performance on the DailyDVS-200 (Wang et al., 2024a) dataset, where EventFlash predicts ac-

Table 5: Action recognition results on processed DailyDVS-200 (Wang et al., 2024a) dataset.

| Methods | Venue | Input Type | Backbone | top-1 acc. (%) |
|---|---|---|---|---|
| **Swin-T** (Liu et al., 2022) | CVPR'22 | Frame | Transformer | 48.06 |
| **GET** (Peng et al., 2023) | ICCV'23 | Event | Transformer | 37.28 |
| **SDT** (Yao et al.) | NeurIPS'24 | Event | Transformer | 35.43 |
| **ESTF** (Wang et al., 2024b) | AAAI'24 | Event | ResNet50 | 24.68 |
| **EventFlash** | Ours* | Event | Qwen2.5 | **48.36** |

tion categories in an open-ended QA setting. Our EventFlash achieves outstanding performance and strong generalization capability.

# 6 CONCLUSION

This paper presents EventFlash, a novel efficient MLLM that leverages spatiotemporal token sparsification to reduce data redundancy and accelerate inference. We also built a large-scale dataset for event stream understanding. The results show that EventFlash achieves a $12.4\times$ improvement in throughput over our baseline (EventFlash-Zero) while maintaining comparable performance. Notably, EventFlash enables long-range event stream processing of up to 1,000 bins compared to only 5 bins in the EventGPT. Our EventFlash serves as an efficient foundational model for event-based vision.

# 7 ACKNOWLEDGMENTS

This work is supported by the National Natural Science Foundation of China (Grant No. 62502317).

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
