# OpenReview forum: "EventFlash: Towards Efficient MLLMs for Event-Based Vision"
_ICLR.cc/2026/Conference — ICLR 2026 Poster_

### Official Review · Reviewer_U2bJ · 2025-10-27

**Soundness:** 3
**Presentation:** 4
**Contribution:** 3
**Rating:** 6
**Confidence:** 4

**Summary:**

This paper proposes an event-based multimodal large model named EventFlash. The method explores reducing data redundancy and accelerating inference for event cameras through spatiotemporal token sparsification. It introduces an adaptive temporal window aggregation module and a sparse density-guided attention module to perform efficient sampling and selection of spatiotemporal information. In addition, the authors construct a large-scale, scene-diverse dataset called EventMind. Experimental results demonstrate that the proposed approach significantly improves throughput while maintaining competitive performance.

**Strengths:**

1. The paper presents a clear motivation. It effectively addresses the high temporal resolution and sparsity characteristics of event camera data, significantly reducing data redundancy and improving the efficiency of large language model (LLM) inference.

2. It provides a scene-driven large-scale event dataset, EventMind, which makes a valuable contribution to the advancement of this field.

3. The experimental results are comprehensive, including images, tables, videos, and code. The writing is fluent, logically structured, and easy to follow.

**Weaknesses:**

1. The authors emphasize the advantages of their method in terms of efficiency and throughput. However, in Table 1, EventFlash-7B does not show a throughput advantage compared with EventGPT-7B. The authors should provide a discussion to clarify this discrepancy.

2. EventGPT-7B was not evaluated on the EventMind dataset. According to Figure 6, EventGPT-7B appears to work on the EventMind dataset, so why are there no quantitative comparison results presented in Table 1? The authors should explain this omission.

3. The application scope seems limited. When constructing the EventMind dataset, the authors used GPT-4o and Qwen-VL-Max to automatically generate annotations. This suggests that similar tasks could potentially be handled using standard RGB cameras combined with LLMs. Therefore, the authors should clarify why event cameras and EventFlash are necessary in this context, and how the advantages of event cameras are effectively leveraged.

**Questions:**

1. When processing the event data, do the authors first stack events into event counts at fixed time intervals, and then aggregate them into W₁, W₂, …, Wₙ using the Adaptive Temporal Window Aggregation module?

2. In Table 1, are the video-based methods also using event data as input? Have these video-based methods been fine-tuned on event data before comparison?

3. Can the proposed Adaptive Temporal Window Aggregation module and Sparse Density-Guided Attention module be applied to other event-based tasks, such as semantic segmentation or optical flow estimation?

---

> ### Author Response · Authors · 2025-11-27
> **Response to reviewer U2bJ (1/2)**
>
> We sincerely thank **Reviewer U2bJ** for the careful reading of our work and positive feedback. We appreciate your recognition of our paper’s clear motivation, the effective handling of event data, the contribution of our large-scale dataset, and the overall clarity of our experiments and writing. Below, we respond to your comments and suggestions in detail.
>
> > **Q1:** *"The authors emphasize the advantages of their method in terms of efficiency and throughput. However, in Table 1, EventFlash-7B does not show a throughput advantage compared with EventGPT-7B. The authors should provide a discussion to clarify this discrepancy."*
>
> **A:** We appreciate the reviewer’s observation. As shown in Table 1, EventGPT attains high raw throughput, but we achieve higher **context-normalized throughput**.
>
> Specifically, to support long-term event stream understanding, we extend the 5-bin event context of EventGPT to the 1000-bin context supported by EventFlash, **a 200× increase**. Although EventGPT-7B reaches 42.2 tokens/s, this relies on an extremely short 5-bin context that loses motion details in long-duration streams. In contrast, EventFlash-7B maintains 24.0 tokens/s with a 1000-bin context, delivering 200× longer coverage with less than a **2× throughput drop**.
>
> Thus, in terms of context-normalized efficiency, EventFlash offers substantially higher **effective throughput** for long-term event understanding.
>
> Excerpted from Table 1 in the main paper:
> | Model          | Max Bins | Throughput |
> |----------------|----------|------------|
> | EventGPT-7B    | 5        | 42.2       |
> | EventFlash-7B  | **1000** | 24.0       |
>
> We thank the reviewer for the suggestion, and we will include this discussion in the camera-ready version of the paper.
>
> > **Q2:** *"EventGPT-7B was not evaluated on the EventMind dataset. According to Figure 6, EventGPT-7B appears to work on the EventMind dataset, so why are there no quantitative comparison results presented in Table 1? The authors should explain this omission."*
>
> **A:** Thank you for the question. EventGPT-7B uses a **fixed 5-bin** partition for all inputs, which works well on short event streams but loses substantial motion details on long sequences. Thus, reporting its quantitative results on EventMind would be misleading. Consistent with Table 1, we only present performance on EventChat-Sub, which focuses on short Event streams. To further illustrate this gap, Figure 6 shows that EventGPT-7B typically yields moment-level descriptions that miss long-range dynamics, whereas EventFlash produces coherent sequence-level interpretations that capture temporal evolution more faithfully.
>
> > **Q3:** *"The application scope seems limited. When constructing the EventMind dataset, the authors used GPT-4o and Qwen-VL-Max to automatically generate annotations. This suggests that similar tasks could potentially be handled using standard RGB cameras combined with LLMs. Therefore, the authors should clarify why event cameras and EventFlash are necessary in this context, and how the advantages of event cameras are effectively leveraged."*
>
> **A:**  We appreciate this question. While GPT-4o and Qwen-VL-Max were used for annotation, the labels were generated from clean, non-degraded RGB videos that are paired with the event streams, rather than from raw event representations.
>
> **High-quality annotation generation**， Because the instruction data were produced offline, we adopted several measures to ensure high-quality annotation, as illustrated in Fig. 7 of the supplementary material.
>
> **For RGB failure application scope**， GPT-based annotation does not imply that RGB+LLMs can solve the tasks under degradation; EventFlash relies on sensing advantages (e.g., low latency, high dynamic range) that RGB fundamentally lacks.
>
> When constructing the instruction set, we adopted multiple quality-assurance measures, including scene priors, data cleaning, and a combination of automated and manual verification, to ensure high-quality training data. However, during inference, the model must confront extreme real-world conditions (e.g., high-speed motion, low-light environments), such as vehicles in tunnels or the instant a bullet moves, where RGB imagery often fails. As a result, the environments encountered during training and inference can differ substantially.

---

> ### Author Response · Authors · 2025-11-27
> **Response to reviewer U2bJ (2/2)**
>
> > **Q4:** *"When processing the event data, do the authors first stack events into event counts at fixed time intervals, and then aggregate them into W₁, W₂, …, Wₙ using the Adaptive Temporal Window Aggregation module?"*
>
> **A:**  In processing the event data, as the reviewer noted, we first partition the raw event stream into fixed-duration bins. However, we do not stack events into count images; instead, we retain the raw event bins. Our ATWA module then adaptively aggregates these raw bins into $W_1, W_2, \dots, W_n$ based on intrinsic event properties such as density and temporal similarity, and the outputs remain raw event bins throughout.
>
> - **Avoids information loss** by not converting events into fixed-interval count frames, thereby preserving the temporal entropy of the event stream.
> - **Reduces redundancy** through adaptive window grouping while maintaining the high temporal resolution inherent to raw event data.
>
> > **Q5:** *"In Table 1, are the video-based methods also using event data as input? Have these video-based methods been fine-tuned on event data before comparison?"*
>
> **A:**  We appreciate the suggestion. Since event-based MLLMs are still in an early research stage, the number of open-source models available for comparison remains very limited (e.g., EventGPT). To better assess long-term event stream understanding, we additionally build a baseline model **EventFlash-zero** for reference (see Table 1 in the main paper). Furthermore, to examine how general video-based MLLMs adapt when event streams are provided in a video-like form, we convert the raw event stream into event videos (i.e., red–blue polarity frames) and include them in our evaluation.
>
> Overall, we understand the reviewer’s concern and provide comparisons with fine-tuned RGB-based MLLMs using event-image representations. We further adopt parameter-efficient LoRA tuning and report results on two representative RGB-based MLLMs (i.e., Qwen2.5-VL and InternVL-2.5).
>
> | Model        | GDC  | FGQA | HAQA | MCQA |
> |--------------|------|------|------|------|
> | Qwen2.5-VL   | 39.6 | 49.4 | 71.6 | 51.4 |
> | InternVL-2.5 | 41.3 | 52.6 | 65.8 | 48.8 |
> | EventFlash   | 46.8 | 61.1 | 84.9 | 60.0 |
>
> Notably, although fine-tuned RGB-based MLLMs show clear improvements, they still fall short of event-based MLLMs trained directly on raw event streams. RGB-based models, even after fine-tuning on event-image representations, inherit strong RGB/video inductive biases and thus adapt to event data only partially, whereas models trained directly from scratch on raw event streams through our three-stage pipeline are better suited to the event modality.
>
> > **Q6:** *"Can the proposed Adaptive Temporal Window Aggregation module and Sparse Density-Guided Attention module be applied to other event-based tasks, such as semantic segmentation or optical flow estimation?"*
>
> **A:**  We thank the reviewer for the insightful comment. The ATWA and SDGA modules are not tied to the LLM framework; they operate on event stream representation and focus on adaptive temporal aggregation and density-guided sparsity. In principle, these mechanisms can be incorporated into other event-based tasks (e.g., semantic segmentation, optical flow) where long-range temporal structure and sparse event patterns are essential, and we view this as a promising extension direction.
>
> Last but not least, we would like to sincerely thank **Reviewer U2bJ** again for the valuable time and constructive feedback provided during this review.

---

> ### Comment · Reviewer_U2bJ · 2025-11-28
>
> Thanks for the authors’ response and the additional experiments. I have no further questions. Just one small suggestion:
>
> if results of EventGPT-7B on EventMind are available, they should be included as well.

---

> > ### Author Response · Authors · 2025-11-28
> > **Thank you for your acknowledgment**
> >
> > Dear Reviewer U2bJ,
> >
> > Thank you for your follow-up and for acknowledging our clarifications on equivalent throughput and application scope.
> >
> > We appreciate your suggestion regarding the EventGPT-7B results on EventMind. We will include them in the camera-ready version of Table 1.
> >
> > Thank you again for your valuable feedback and consideration.
> >
> > Best regards,
> > The Authors of Submission 15105

---

### Official Review · Reviewer_aCxZ · 2025-11-01

**Soundness:** 2
**Presentation:** 2
**Contribution:** 2
**Rating:** 4
**Confidence:** 5

**Summary:**

This paper presents an efficient MLLM for event-based vision tasks. The authors claim that this is the first efficient MLLM to explore spatio-temporal token sparsification, aiming to reduce redundancy, accelerate inference, and enable long-range event stream understanding. To address temporal inefficiency, the authors propose a two-stage density-guided Adaptive Temporal Window Aggregation (ATWA) module; to address spatial redundancy, they introduce a Sparse Density-Guided Attention (SDGA) module. To demonstrate performance on long-range event tasks, the authors also generate a large dataset containing 500k events. Experiments are conducted to evaluate the proposed framework on the self-generated dataset across seven tasks: motion captioning, event question answering (Event QA), human action QA, multiple-choice QA (MCQA), simple captioning, fine-grained QA (FGQA), and scene captioning. The authors report throughput performance compared to state-of-the-art (SOTA) methods using an A100 GPU. Some qualitative results are also provided.

**Strengths:**

1. A novel self-generated dataset that could be useful for the event-based MLLM research community.
2. Two interesting modules appear to help improve throughput while achieving comparable accuracy.
3. Some qualitative analyses are provided.

**Weaknesses:**

1. The experimental comparison to actual SOTA MLLMs (e.g., Qwen2.5-VL, InternVL, LLaVA-v1.6) is critically undermined by the paper's failure to state whether these baselines were finetuned on event data. This strongly suggests an unfair comparison against zero-shot models, rendering the performance results unreliable.
2. The reliance on "RGB-style" (image-like) representations is a significant limitation. The method's effectiveness is never validated against true native event representations (e.g., point clouds, graphs), which would be necessary to prove its value for event-based vision.
3. The framework appears to be an incremental application of common MLLM techniques (e.g., standard token sparsification, three-stage training) to a pre-processed, image-like data format, rather than a fundamental innovation for "raw event streams."
4. The paper's second major contribution, the EventMind dataset, is undermined by critical concerns. Significant doubts exist regarding the quality of annotations generated by GPT-4o, especially given its visual token limits and the challenge of processing long event sequences. The "human checking" and validation process lacks sufficient detail to be verifiable.
5. The paper's headline claim of a "12.4× throughput improvement" is highly misleading. This gain is measured against "EventFlash-Zero," an internal baseline, not against established SOTA models.
6. Benchmarking is incomplete, omitting comparisons against key models like EventGPT-13B and failing to report results on established metrics from prior work (e.g., EventGPT's VQA setting).
7. A fundamental contradiction exists between the paper's premise and its implementation. The paper claims to explore a "sparsification strategy for raw event streams," but the methodology reveals it operates on "image-based tokens" from a standard ViT encoder.

**Questions:**

See weakness. I am frustrated that the author did not make corresponding revisions to the submitted paper based on the NeurIPS review comments. This work has so many limitation, although its packaging is good. I think the event community need the solid and usefull MLLM-based works, not limited in the Line MLLM+some existing techinuqes+datasets''.

---

> ### Author Response · Authors · 2025-11-27
> **Response to reviewer aCxZ (1/3)**
>
> We sincerely thank the reviewer for carefully reading our work and for providing many constructive suggestions. We would like to clarify that EventFlash is designed to investigate **efficient sparsification** within existing event-based MLLM architectures (e.g., EventGPT, EventVL), aiming to address the **high latency** and **low throughput** issues caused by dense event representations when processing **long-term event streams**. Our intention is not to propose an entirely new heterogeneous event architecture.
>
> Moreover, the field currently lacks large-scale, high-diversity event datasets that could support training a fully heterogeneous and highly capable event encoder from scratch. While this remains an important and promising research direction, current data limitations make such training challenging in practice.
>
> Overall, we believe that such step-by-step progress will eventually lead to native asynchronous event-based MLLMs, potentially leveraging representations such as graphs or point clouds. However, our EventFlash focuses specifically on efficient sparsification. We also plan to explore these native asynchronous architectures in future work.
>
> > **Q1:** *"RGB-based MLLM finetuned on event stream"*
>
> **A:**  We appreciate the suggestion. We include video-based MLLMs in zero-shot form not to highlight a performance advantage of EventFlash, but rather to illustrate that current video-based MLLMs are not directly suitable for event stream understanding due to the substantial modality gap between RGB videos and raw event data. Furthermore, to examine how general video-based MLLMs adapt when event streams are provided in a video-like form, we convert the raw event stream into event videos (i.e., red–blue polarity frames). Following common practice, we adopt an FPS = 1 video-like setting and include these representations in our evaluation.
>
> Overall, we understand the reviewer’s concern and provide comparisons with **fine-tuned RGB-based MLLMs** using event-image representations. We further adopt parameter-efficient LoRA tuning and report results on two representative RGB-based MLLMs (i.e., Qwen2.5-VL and InternVL-2.5).
> | Model          | GDC  | FGQA | HAQA | MCQA |
> |----------------|------|------|------|------|
> | Qwen2.5-VL     | 39.6 | 49.4 | 71.6 | 51.4 |
> | InternVL-2.5   | 41.3 | 52.6 | 65.8 | 48.8 |
> | EventFlash | 46.8 | 61.1 | 84.9 | 60.0 |
>
> Notably, although RGB-based MLLMs, even after fine-tuning on event-image representations, still inherit **inductive biases** from their RGB/video pretraining. As a result, they adapt to event data only partially, whereas models trained directly **from scratch** on raw event streams through our three-stage pipeline are better suited to the event modality.
>
> > **Q2& Q3:** *"using an RGB-like representation instead of the raw event representation（i.e., point, graph）"*
>
> **A:** We fully agree with the reviewer that asynchronous representations such as graphs and point clouds can more faithfully preserve the intrinsic structure of raw event data. However, current encoders for these representations remain constrained in practice: they lack large-scale pretraining, rely on small task-specific datasets, and typically underperform in both efficiency and generalization. As a result, their features are difficult to align with LLMs, which is why existing event–language models (e.g., EventCLIP, EventBind) continue to use image-like representations.
>
> Given these constraints, adopting image-like event representations in our work is a pragmatic and effective choice for demonstrating how LLMs’ world knowledge can benefit event understanding, while enabling favorable performance–efficiency trade-offs on raw event streams.
> Nevertheless, we agree that fully asynchronous event-based  MLLMs represent an important and promising future direction, which will require advances in large-scale dataset construction and encoder pretraining. We plan to explore this native asynchronous form in future work.

---

> ### Author Response · Authors · 2025-11-27
> **Response to reviewer aCxZ (2/3)**
>
> > **Q4:** *"The reliability of GPT-4o–generated annotations and the GPT-4o visual token limit."*
>
> **A:**  We appreciate this question. While GPT-4o was used for annotation, the labels were generated from clean, non-degraded RGB videos that are paired with the event streams, rather than from raw event representations.
>
> **GPT-4o visual token limit.** GPT-4o supports a 128k context window. As stated in the main paper (line 151–152), the temporal span of each long event stream is **below 20,000 ms (~20 s)**. With a typical sampling rate of 1-4 FPS, a 20-second RGB video max contains 80 frames. Each frame is encoded into roughly 170 visual tokens, resulting in a total of 170 × 80 = 13.6k tokens, which is far below the **128k context window limit**. Thus, GPT-4o can safely process all frames without exceeding the context length.
>
> **Annotation quality & Human checking process.** Figure 7 in the supplementary material details our instruction-generation and quality-control pipeline. We design a two-stage prompt-checking process: (i) prompts are crafted to emphasize motion dynamics while avoiding static or color-dependent cues irrelevant to event streams, and (ii) a lightweight MLLM (Qwen2.5-VL-3B) is used to extract structured metadata (e.g., action, object) for pre-validation. If the metadata matches the GPT-4o-generated instruction, the sample is retained; otherwise, it is manually inspected. In fact, similar LLM-generated and quality-controlled data pipelines have also been used in other modality-specific domains, including infrared imagery (IRGPT) and remote sensing (EarthGPT).
>
> > **Q5:** *"The paper's headline claim of a "12.4× throughput improvement" is highly misleading. This gain is measured against "EventFlash-Zero," an internal baseline, not against established SOTA models."*
>
> **A:**  We appreciate this concern, and we agree that comparing against more event-based MLLMs would be preferable. Yet given their lack of public availability, adopting EventFlash-Zero as the baseline is a reasonable and practical choice under these constraints.
>
> **Model Availability** — Current event-based MLLMs (e.g., EventVL, LLaFEA, LET-US) are not open-sourced, so throughput cannot be measured; this is a practical limitation, not an intentional omission.
>
> **Baseline Rationale** — In the early exploratory stage of a field, when reproducible open-source baselines are lacking, adopting a classic architecture as a reference model is a common and justified practice (e.g., LLaVA, Flamingo). Accordingly, EventFlash-Zero uses a standard LLaVA-like design to provide a fair and transparent baseline for evaluating the performance gains brought by our sparsification strategy.
>
> **Relative Gain** — The **12.4× improvement** reflects the relative gain within a controlled and fully reproducible setting, rather than an absolute comparison against unavailable SOTA implementations.

---

> ### Author Response · Authors · 2025-11-27
> **Response to reviewer aCxZ (3/3)**
>
> > **Q6:** *"Benchmarking is incomplete, omitting comparisons against key models like EventGPT-13B and failing to report results on established metrics from prior work (e.g., EventGPT's VQA setting)."*
>
> **A:**  Thank you for the question. EventFlash is primarily designed to investigate sparsity acceleration for event-based MLLMs, rather than to propose a performance-oriented SOTA model.
>
> **Comparison at the same model size.** To fairly evaluate the efficiency and throughput improvements introduced by EventFlash’s sparsification strategy, we compare models at the same parameter scale. As shown in Table 1, all throughput results are reported relative to models of the same size. Naturally, larger models exhibit lower throughput on identical hardware. If additional event-based MLLMs become open-sourced in the future, we will include them to further validate our sparsification strategy.
>
> **EventGPT’s VQA setting.** The EventChat-Sub dataset is exactly the subset released by EventGPT on HuggingFace, and we follow their evaluation setup when reporting results in Table 1. This is explicitly clarified in lines 325–327 of the main paper.
>
> In summary, our evaluation focuses on fair, same-size comparisons to measure efficiency and throughput gains, across both our newly constructed EventMind benchmark and the publicly available EventChat-Sub dataset.
>
> > **Q7:** *"A fundamental contradiction exists between the paper's premise and its implementation. The paper claims to explore a "sparsification strategy for raw event streams," but the methodology reveals it operates on "image-based tokens" from a standard ViT encoder."*
>
> **A:**  Thank you for the question. Although the final input to the MLLM is an image-like representation, both sparsification modules in Figure 2 operate directly on the raw event stream and are explicitly guided by its intrinsic properties.
>
> **ATWA (temporal sparsification).** As shown in Figure 2, ATWA does not use any image-like intermediate; it processes the raw event stream based on density and temporal similarity to adaptively form multi-resolution event bins, effectively reducing temporal redundancy.
>
> **SDGA (spatial sparsification).** Before feature extraction, SDGA computes raw event statistics such as density cues to guide the pruning of spatial tokens. Thus, SDGA performs raw-event-driven spatial pruning, fundamentally different from RGB-style pruning methods.
>
> We fully agree with the reviewer that **asynchronous event-driven MLLMs** (e.g., point, graph) represent an important future direction. However, we believe that raw event representations still face several practical limitations, including the lack of large-scale datasets and the absence of efficient encoding structures. Moreover, even if the event stream is processed asynchronously in the early stages, its features must ultimately be converted into **conventional embeddings** before entering the LLM backbone. We sincerely hope the reviewer understands this constraint. The goal of EventFlash is to address the **high latency** and **low throughput** issues that arise specifically from dense event representations, rather than to design a fully asynchronous architecture.
>
> Last but not least, we would like to sincerely thank **Reviewer aCxZ** again for the valuable time and constructive feedback provided during this review.

---

### Official Review · Reviewer_UMdA · 2025-11-01

**Soundness:** 3
**Presentation:** 3
**Contribution:** 3
**Rating:** 6
**Confidence:** 2

**Summary:**

This paper addresses the inefficiency of applying dense, frame-like processing to sparse event-based MLLMs. It makes two main contributions: (1) EventMind, a new 500k-sample instruction dataset for event vision, enabling a short-to-long curriculum learning strategy; and (2) EventFlash, an efficient MLLM using adaptive temporal (ATWA) and sparse spatial (SDGA) token sparsification(2). Experiments show EventFlash achieves a 12.4x throughput gain over its non-sparse baseline and can process much longer event sequences (1,000 bins).

**Strengths:**

1. The creation of the 500k-sample EventMind dataset is a major contribution that addresses a critical resource gap for training and benchmarking event-based MLLMs.
2. The paper targets the correct bottleneck: the inefficiency of applying dense methods to sparse data. The proposed spatiotemporal sparsification modules (ATWA and SDGA) are an intuitive and direct solution to this problem.

**Weaknesses:**

1. Insufficient comparison to SOTA event-based models: The paper fails to benchmark EventFlash against its direct competitors. On its new EventMind dataset, it only compares against frame-based models (Table 1). On the existing EventChat-Sub dataset, it only compares against EventGPT, omitting other SOTA event models like EventVL mentioned in the related work. This makes the SOTA performance claims unsubstantiated.
2. Missing Key Methodological Ablations: The core Adaptive Temporal Window Aggregation (ATWA) module is a complex two-stage process (a spike-based merge followed by a semantic-based merge. However, the ablation study (Table 2) only validates the entire "+T" (Temporal) block at once. It never justifies the necessity of this complex two-stage design over a simpler, single-stage alternative.

**Questions:**

See weaknesses above.

---

> ### Author Response · Authors · 2025-11-27
> **Response to reviewer UMdA**
>
> We sincerely thank **Reviewer UMdA** for the careful reading of our work and positive feedback. We appreciate your recognition of our paper’s clear motivation and the contribution of our large-scale dataset, and the overall clarity of our experiments and writing. Below, we respond to your comments and suggestions in detail.
>
> > **Q1:** *"Insufficient comparison to SOTA event-based models: The paper fails to benchmark EventFlash against its direct competitors. On its new EventMind dataset, it only compares against frame-based models (Table 1). On the existing EventChat-Sub dataset, it only compares against EventGPT, omitting other SOTA event models like EventVL mentioned in the related work. This makes the SOTA performance claims unsubstantiated."*
>
> **A:**  We appreciate this concern. Event-based MLLMs are still in an early stage of exploration, and most existing models are not publicly accessible. EventGPT is currently the only open-source event-based MLLM, and we have already included it as a baseline in Table 1. In addition, to more rigorously demonstrate the sparsity acceleration of EventFlash, we constructed a LLaVA-like architecture, EventFlash-Zero, for comparison. Experiments show that EventFlash achieves a **12.4×** throughput improvement. Specifically:
>
> **Model Availability** — Current event-based MLLMs (e.g., EventVL, LLaFEA, LET-US) are not open-sourced, so throughput cannot be measured; this is a practical limitation, not an intentional omission.
>
> **Baseline Rationale** — In the early exploratory stage of a field, when reproducible open-source baselines are lacking, adopting a classic architecture as a reference model is a common and justified practice (e.g., LLaVA, Flamingo). Accordingly, EventFlash-Zero uses a standard LLaVA-like design to provide a fair and transparent baseline for evaluating the performance gains brought by our sparsification strategy.
>
> We acknowledge that including more event-based MLLMs would further validate the acceleration advantages of EventFlash relative to dense processing models. Should these models become available in the future, we plan to incorporate them in subsequent versions of EventFlash and discuss the results in the revised manuscript.
>
> > **Q2:** *"Missing Key Methodological Ablations: The core Adaptive Temporal Window Aggregation (ATWA) module is a complex two-stage process, a spike-based merge followed by a semantic-based merge. However, the ablation study (Table 2) only validates the entire "+T" (Temporal) block at once. It never justifies the necessity of this complex two-stage design over a simpler, single-stage alternative."*
>
> **A:** Thank you for this question. We would like to clarify that, for the ATWA module, the two stages serve different and sequential roles: the first stage operates on raw asynchronous spikes to reduce low-level temporal redundancy before any encoding, while the second stage performs semantic-level merging on the structured meta-windows produced by Stage 1. Since Stage 2 requires the output of Stage 1, and Stage 1 cannot leverage semantic embeddings that only emerge after Stage 2, the two stages rely on different representations and therefore cannot be cleanly collapsed or ablated in isolation.
>
> Regarding ablations, Table 2 already evaluates the temporal and spatial sparsification components separately. Temporal sparsification alone improves throughput from **2.3 to 14.0 tokens/s**, and spatial sparsification alone achieves 5.3 tokens/s. These results directly quantify the contributions of the two core sparsification dimensions and demonstrate that both temporal and spatial mechanisms provide meaningful gains.
>
> **Excerpted from Table 2 in the main paper:**
>
> | Model    | Token/s     | GDC  | FGQA | HAQA | MCQA |
> |----------|-------------|------|------|------|------|
> | Baseline | 2.3         | 45.3 | 60.4 | 85.0 | 58.2 |
> | A (S+)   | 5.3  (+2.3×) | 46.3 | 61.2 | 85.1 | 59.6 |
> | B (T+)   | 14.0 (+6.1×) | 47.1 | 60.6 | 83.8 | 60.3 |
>
> We will further clarify these points in the revised manuscript to better convey the sequential nature and contributions of ATWA.
>
> Last but not least, we would like to sincerely thank **Reviewer UMdA** again for the valuable time and constructive feedback provided during this review.

---

> ### Comment · Area_Chair_BANm · 2025-12-02
>
> Dear Authors,
>
> Are you going to make data and code open source?

---

> > ### Author Response · Authors · 2025-12-02
> >
> > **Dear Area Chair BANm,**
> >
> > Thank you for your time in serving as the AC for our paper. All resources will be fully organized and made publicly available before the ***camera-ready***.
> >
> > EventMind is one of the **major contributions** of our work. Given the current lack of high-quality event–text datasets in the community, we are prioritizing its preparation and will release it promptly on HuggingFace. To further enhance reproducibility, we also commit to releasing the complete implementation of EventFlash on GitHub.
> >
> > **Release Timeline**
> >
> > **- By February 2026:** Public release of the EventFlash codebase on GitHub.
> >
> > **- By March 2026:** Completion and release of the EventMind dataset on HuggingFace.
> >
> >
> > Best regards,
> >
> > The Authors of Submission 15105

---

### Author Response · Authors · 2025-12-02
**Summary of Responses**

**Dear ACs and SACs,**

We sincerely thank you for handling our submission and the reviewers for their constructive feedback.

We are encouraged by the **recognition** of our contributions, including:

- Reviewer **UMdA** highlights this work for its “major contribution” in creating EventMind and for targeting the “correct bottleneck”.

- Reviewer **aCxZ** recognizes the “useful dataset”, the “throughput-improving modules”, and the “helpful qualitative analyses”.

- Reviewer **U2bJ** commends this work for its “clear motivation”, its “effective treatment of event-data sparsity and timing”, and its “comprehensive experiments”.

In response to the reviewers’ insightful comments, we have made the following **clarifications and improvements**:

- Reviewer **UMdA’s** main question focuses on comparisons with other event-based MLLMs and the discussion of ablation studies. We clarified the practical limitation that most existing event-based MLLMs are not publicly available, and explained that our baseline model, EventFlash-Zero, was constructed following a reasonable design for efficiency analysis. We also clarified that the two stages of ATWA form a sequential and interdependent process, which cannot be cleanly separated.

- Reviewer **aCxZ** primarily raised questions regarding comparisons with zero-shot RGB-based MLLMs and the use of asynchronous event representations (e.g., point, graph). In response, we included additional experiments comparing our method with fine-tuned RGB-based MLLMs. We also acknowledged that asynchronous event representations constitute an important future direction for event-based MLLMs. Moreover, EventFlash is designed to achieve efficient sparsification within existing event-based MLLMs to alleviate latency and throughput bottlenecks caused by dense event representations, rather than to introduce a new heterogeneous architecture.

- Reviewer **U2bJ** primarily raised questions regarding the throughput comparison with EventGPT, performance on EventMind, and the applicability relative to RGB-based MLLMs. We clarified that EventFlash achieves higher effective throughput, enabling nearly a 200× expansion in event bins while requiring less than a 2× increase in computational cost. We also explained that EventGPT cannot effectively handle long-term event streams in EventMind due to its fixed-window constraint. In addition, we included supplementary experiments comparing our method with fine-tuned RGB-based MLLMs.

Thank you again for your time and consideration!

Warmest regards,

The Authors of Submission 15105

---

### Meta-Review · Area_Chair_BANm · 2025-12-07

**Summary:**

This paper proposes an efficient event-based multimodal large language model using efficient sparsification to alleviate latency and throughput bottlenecks caused by dense event representation. In addition, the authors build a 500k-sample event-instruction dataset intended to support long-term event stream understanding. Experiments report throughput improvement over a dense baseline and demonstrate results across multiple tasks (captioning, QA variants).

Reviewers recognise the creation of dataset and well motivation as an effective treatment of event-data sparsity and timing. One of main concerns from reviewers is lack of comparisons with other event-based MLLMs due to no other public available source code. Since authors have listed the time line for releasing the dataset and code, the submission is recommend for acceptance.

**Reviewer Concerns:**

A primary concern across reviews. Several key SOTA event-based MLLMs are missing or compared under unclear training settings. Since most existing event-based MLLMs are not publicly available,this submission whose code and dataset will be released is an important contribution.

Reviewer asked to compare with zero-shot RGB-based MLLMs and the use of asynchronous event representations. Rebuttal provides additional experiments comparing our method with fine-tuned RGB-based MLLMs.

Reviewer U2bJ raised questions on comparison with EventGPT, performance on EventMind. Authors have explained them.

**Reviewer Scores:**

Reviewer UMdA and  U2bJ  may increase or remain unchanged.

Reviewer aCxZ might increase the score to acceptance.

---

### Decision · Program_Chairs · 2026-01-26

Accept (Poster)